# Autophagy Function and Dysfunction: Potential Drugs as Anti-Cancer Therapy

**DOI:** 10.3390/cancers11101465

**Published:** 2019-09-29

**Authors:** Francesca Cuomo, Lucia Altucci, Gilda Cobellis

**Affiliations:** Department of Precision Medicine, University of Campania “L. Vanvitelli”, via L. De Crecchio, 7, 80138 Naples, Italy; francesca.cuomo@unicampania.it

**Keywords:** autophagy, cancer, apoptosis, *chloroquine*, mTOR inhibitors, drugs

## Abstract

Autophagy is a highly conserved catabolic and energy-generating process that facilitates the degradation of damaged organelles or intracellular components, providing cells with components for the synthesis of new ones. Autophagy acts as a quality control system, and has a pro-survival role. The imbalance of this process is associated with apoptosis, which is a “positive” and desired biological choice in some circumstances. Autophagy dysfunction is associated with several diseases, including neurodegenerative disorders, cardiomyopathy, diabetes, liver disease, autoimmune diseases, and cancer. Here, we provide an overview of the regulatory mechanisms underlying autophagy, with a particular focus on cancer and the autophagy-targeting drugs currently approved for use in the treatment of solid and non-solid malignancies.

## 1. Introduction

Autophagy is a highly conserved catabolic and energy-generating process that facilitates the degradation of intracellular materials or damaged organelles, providing cells with monomeric building blocks that can be released back into the cytoplasm for the synthesis of new components, and the production of energy [1]. Autophagy, derived from the Ancient Greek *autóphagos* meaning “self-eating” coined by Christian de Duve in 1963, is induced in response to different stressors, such as amino acid starvation, glucose deprivation, oxygen deficiency, growth factor withdrawal, or DNA damage, allowing cells to compensate for the insult [1]. In 2016, Yoshinori Ohsumi was awarded the Nobel Prize in Physiology or Medicine for his discoveries on autophagy, elucidating the mechanisms of this process in healthy conditions and highlighting its pro-survival role. 

The deregulation of autophagy is often associated with cell death, which is a “positive” and desired biological choice in some circumstances, or with the genesis of pathological states. Due to its involvement in many disorders, autophagy has become a major target for drug discovery. Recently, great attention has focused on the role of autophagy in cancer development and progression [2,3]. Here, we provide an overview of the regulatory mechanisms underlying autophagy and autophagy-targeting drugs. We also discuss the results of clinical trials investigating the effects of molecules affecting autophagy in cancer treatment.

## 2. Autophagy and Regulatory Machinery

Three different forms of autophagy have been described: microautophagy, macroautophagy, and chaperone-mediated autophagy (CMA) [4]. Microautophagy involves the direct engulfment of cytoplasmic material at lysosomes or late endosomes. CMA is a selective process: specific substrates containing the KFERQ motif are identified by the chaperone heat shock cognate protein 70, which transports the protein cargo to the lysosomal membrane, where it is linked to the cytosolic tail of lysosome-associated membrane protein type 2A and degraded [4].

Macroautophagy (hereafter referred to autophagy) is a multistep process involving:-the formation of a cytoplasmic double-membrane structure called a phagophore-phagophore elongation, maturation, and closure, followed by the engulfment of long-lived cytosolic proteins and/or damaged organelles (mitochondria, ribosomes, etc.) with the formation of double-membrane vesicles, which are known as autophagosomes-the fusion of autophagosomes with lysosomes, forming the autolysosome, which contains acid hydrolases and where degradation of the cargo occurs (Figure 1) [4]. 

Evolutionarily conserved genes called autophagy-related genes (ATGs) are the main players in this process. Under stress, cells activate the ULK1 complex, consisting of ULK1/2, Atg13, Atg101, and focal adhesion kinase family-interacting protein of 200 kDa (FIP200) [1,3], which in turn phosphorylates the Beclin complex (vacuolar protein sorting [Vps]34, Vps15, Atg14, and Beclin-1), thereby initiating phagophore formation [4].

The phagophore begins to engulf cytosolic macromolecules via the coordinated action of ubiquitin-like conjugation systems (Atg5-Atg12 ubiquitin conjugation enzymes, Atg7, and Ub-activating enzyme Atg16, via LC3–phosphatidylethanolamine (PE) conjugation [LC3-II]). The autophagic cargo is recognized by SQSTM1/p62, which is a protein that is able to sequester polyubiquitinated proteins, linking them to Atg8/LC3II complex on autophagosomal membranes [5]. During this process, p62 forms cytoplasmic “bodies”, non-membrane structures that compartmentalize ubiquitinated macromolecules before LC3 interaction [6]. The autophagosomes fuse with endosomes and late lysosomal in which lysosomal acid hydrolases degrade the cargo, and the resulting small molecules (nucleotides, amino acids, fatty acids, and sugars) are then transported back to the cytosol for reuse through lysosomal membrane permeases (Figure 1) [4].

Lysosomes were originally thought to be static organelles involved in the clearance of waste material; this idea was challenged by the discovery of autophagy, which exerts a tight control on cellular homeostasis. The transcription factor EB (TFEB) and transcription factor E3 (TFE3) are regulators of autophagy and lysosomal biogenesis. Under nutrient-rich conditions, TFEB is sequestered in the cytosol in a phosphorylated state. The mechanistic target of rapamycin complex 1 (mTORC1) negatively regulates TFEB activity in a nutrient-dependent manner, contributing to the switch between anabolic and catabolic pathways [7]. Nutrient deprivation stimulates ULK1–Beclin complexes to initiate autophagy [8]. Then, TFEB and TFE3 are dephosphorylated and translocate to the nucleus, activating lysosomal genes. In these conditions, low ATP/high AMP levels occurs, activating AMP kinase (AMPK) to minimize ATP usage and activating autophagy. In addition, AMPK phosphorylates and activates p27^kip1^, which is a cyclin-dependent kinase inhibitor leading to cell cycle arrest, preventing cells from undergoing apoptotic death and switching to autophagy as a pro-survival mechanism in stress conditions [9]. In contrast, an excess of nutrients leads to AKT activation, high mTORC1 activity, the phosphorylation of TFEB and TFE3, and the subsequent inhibition of autophagy (Figure 2) [9].

mTORC1 is associated with endosomal and lysosomal membranes, and its activity is mainly regulated by the positive regulatory-associated protein of mTOR (RAPTOR) and the negative regulator PRAS40 [10]. Once activated by AKT, mTORC1, either by the dissociation of PRAS40 from RAPTOR or the inhibition of TSC1/2 complex formation, releasing RHEB, binds via Rag-GTPase to lysosomes, where mTORC1 activates protein synthesis modulators (4EBP1 and S6K1) and ribosome genesis, promoting G1-S phase transition [11].

The regulation of autophagy is also mediated by mTORC2, which is a less well characterized pathway that inhibits apoptosis in response to growth factor stimulation, and regulates cell migration and cytoskeletal rebuilding [4]. mTORC2 localizes to plasma membranes, where through its scaffolding protein RICTOR binds mTOR, and mammalian stress-activated protein kinase interacting protein 1 (mSIN1) [12].

Dysregulation of the autophagic process is associated with a wide range of human diseases, including neurodegenerative, lysosomal storage, bone, and metabolic disorders, and cancer [13,14]. Current knowledge suggests that this process is regulated at epigenetic, transcriptional, and post-translational levels [15]. A better understanding of autophagic molecular mechanisms and regulation under different conditions may help to identify potential novel targets for the treatment of several human diseases. In this review, we focus on cancer.

## 3. Role of Autophagy in Cancer

Cancer is generally characterized by a genetic and metabolic imbalance that allows abnormal cell growth. Autophagy is considered a “double-edged sword” due to its opposite effects in cancer, where it can both protect against the development of a tumor by depriving cells of nutrients and energy, or support tumor progression by conferring a metabolic advantage [16].

The failure of autophagy at any step leads to upregulated or decreased autophagic flux, which affects tumor growth over time. Notably, increased proliferation in pancreatic cancer cells is associated with TFEB overactivation, providing an excess of nutrients obtained by the glycogen synthase kinase-3 (GSK3) inhibitors [17]. TFEB and TFE3 chromosomal translocations were found in clear cell renal cell carcinoma (RCC) [18], and tumor progression is enabled by an induction of cyclin D2, cyclin D3, and p21 [7,19]. The enhanced motility of non-small cell lung cancer (NSCLC) cells and cancer aggressiveness was linked to increased TFEB expression [20], which promotes cancer cell proliferation and/or the induction of anti-apoptotic genes, such as *BCL2* and *BIRC7* in melanoma [21], or fuels cell metabolism and supports tumor progression in pancreatic cancer [22]. The rapid growth of cancer cells benefits from the overexpression of SQSTM1/p62, due to constitutive nuclear factor kappa-light-chain-enhancer of activated B cells (NF-kB) activation [23]. SQSTM1/p62 can be inactivated by phosphorylation under oxidative stress conditions, leading to a loss of autophagic activation and subsequent nuclear factor erythroid 2-related factor 2 (Nrf2) activation [24]. Nrf2 regulates the expression of genes encoding detoxifying enzymes, stress response proteins, and metabolic enzymes, thus eliminating ROS-induced metabolic products, and providing nutrients required for cancer cell survival [24].

Recently, the conditional knock-out of *p62* in adipocytes of prostate cancer model (TRAMP^+^) was found to suppress the energy-consuming pathways in these cells, increasing nutrient supply to surrounding cancer cells [25].

Together, these studies indicate a multifaceted and complex role for p62.

The loss of BECN1 is a frequent event in many human tumors, including breast, ovarian, and prostate cancer, and murine *Beclin-1* knock-out studies confirmed the development of spontaneous tumors in mice [26]. The allelic loss of proteins interacting with Beclin-1, such as UV radiation resistance-associated gene protein (UVRAG) and Bax-interacting factor-1, is also common in breast, gastric, colon, bladder, and prostate cancer [27]. *Bif-1* knock-out led to autophagic impairment and an increase in the rate of tumor formation, while UVRAG mutations reduced autophagy with an increase in colorectal cancer (CRC) cells [28]. The deletion of *Atg5* in mice led to chronic inflammation, chronic liver damage, and subsequent tumor development [29]. *Atg7* knock-out also resulted in an increase in liver tumors, and Atg4 deficiency led to the development of fibrosarcoma [29,30].

TSC is an autosomal dominant human syndrome that causes spontaneous tumor formation in multiple organs, caused by inactivating mutations in genes encoding TSC1/2, which are negative regulators of mTORC1 [31].

Thus, mutations in most autophagy-associated genes promote tumorigenesis by altered autophagic flux. However, several cancers display high levels of autophagy, suggesting that other events activate autophagy to sustain cell transformation. Cancer cell proliferation in RAS- and/or RAF-mutated tumors is maintained by increased autophagy [32,33]. Therefore, the involvement of autophagy in tumorigenesis is highly controversial, due to its evident crosstalk with cell death programs (i.e., apoptosis, necrosis).

## 4. Autophagy and Apoptosis

Apoptosis and autophagy are both vital cellular processes, and crosstalk between these two events is complex and influences cell homeostasis. Autophagy was found to precede, and thus inhibit, apoptosis in low non-lethal stress conditions, acting in a pro-survival manner. Persistent and high levels of stress lead to the activation of apoptosis suppressing autophagy [34]. The caspase-dependent cleavage of Atg3, Atg4, Atg5, and Beclin1 produces fragments with a pro-apoptotic function that were shown to translocate to mitochondria and stimulate mitochondrial outer membrane permeabilization (MOMP) [35]. In these conditions, cells shift to glycolytic metabolism, arresting the oxidative respiration and utilizing glutamine as fuel, for which autophagy is required [36,37]. Apoptosis, on the contrary, inhibits Beclin and the autophagosome formation, which is a process rescued by anti-apoptotic Bcl-xL protein [36,37,38]. Caspase-10 was found to have a cytoprotective role, since it induces autophagy by stimulating the degradation of Bcl-2-associated transcription factor 1 (BCLAF) or other Atg proteins. Atg12 interacted with anti-apoptotic Bcl-2 and Mcl-1, stimulating apoptosis [39], while Atg7 induced apoptosis by promoting membrane permeabilization in response to lysosomal photodamage [40]. Furthermore, cFLIP was shown to inhibit autophagy by competitive binding to Atg3, therefore preventing the processing of LC3 [41].

Mitochondria play a central role in crosstalk between autophagy and apoptosis, since MOMP is essential to activate the intrinsic apoptotic response, whereas damaged mitochondria stimulate mitophagy through their degradation and the subsequent inhibition of apoptosis [42]. In Ras-transformed cells, mitophagy takes part in the shift to glycolysis, and hence is required to maintain metabolic homeostasis, inhibiting the respiration [42]. Mitochondria damage is also associated with the activation of necrosis, and is considered a master regulator of ˝danger˝ signaling [43].

A high-frequency mutation rate of *p53* was observed in several cancer cells, especially in NSCLC, suggesting its possible role in regulating both autophagy and apoptosis [44]. When p53 is cytosolic, it inhibits autophagy by interacting with FIP200, whereas when phosphorylated, it translocates into the nucleus and activates gene expression programs regulating cell cycle arrest, autophagy, and apoptosis. Cytosolic p53 is also able to translocate into mitochondria causing MOMP, leading to either mitophagy or apoptosis in response to low or high levels of stress, respectively [45]. In contrast, p73, a member of the same protein family, was shown to activate the mTOR signaling pathway and thus inhibit autophagy [46]. Curiously, the p53 target damage-regulated autophagy modulator (DRAM) is inactivated in several tumors, causing a switch from autophagy to apoptosis [47]. However, the molecular mechanisms underlying the regulation of this and other p53 targets need to be further investigated in order to understand how p53 controls autophagy and apoptosis [47].

BH3-only proteins are also able to induce both autophagy and apoptosis. Apart from their well-known role in apoptosis, they are also able to interrupt the inhibitory role of Bcl-2 proteins on Beclin 1 and induce autophagy. Serine/threonine kinases are also reported to regulate both processes, since active death-associated protein kinase and c-Jun N-Terminal Kinase (JNK), as well as inhibiting AKT, simultaneously induce apoptosis and autophagy [46].

It is becoming clear that the interplay between autophagy and apoptosis is complex and cell-dependent. For example, autophagosome accumulation may stimulate MOMP and subsequently the intrinsic apoptotic pathway [46]. In addition, the autophagosome membrane, as well as cytosolic p62 accumulation [48], generates a platform for caspase-8 activation [49]. When cell damage can be repaired, mitophagy is activated and dysfunctional mitochondria are sequestered, releasing apoptotic and anti-apoptotic proteins, and allowing a switch to glycolysis to meet energy requirements. When cell damage is too extensive, metabolic balance cannot be maintained and apoptosis is initiated, at least in neurodegeneration [50].

## 5. Should We Try to Enhance or Inhibit Autophagy in Cancer?

### 5.1. Autophagy Inhibitors

Altering autophagic flux is an exploitative cancer therapy, either downregulating or upregulating it. To date, three different approaches have been explored: inhibiting lysosome and autophagy cargo degradation, blocking essential components of autophagy machinery, and repressing mitochondrial respiration [51], although the mechanisms underlying mitochondrial respiration are still debated.

The few autophagy inhibitors developed to date display therapeutic potential in counteracting the growth of tumor cells; these inhibitors target nucleation and extension of the phagophore or block the endosomal/lysosomal acidification process (Table 1).

Chloroquine (CQ) and hydroxychloroquine (HCQ) are two commercially available drugs under investigation for the treatment of cancer. CQ has long been used to treat malaria caused by *Plasmodium vivax*, *ovale*, *malariae* [52], and *falciparum,* while HCQ, a less toxic metabolite of CQ, is used to treat rheumatic diseases such as systemic lupus erythematosus, rheumatoid arthritis, juvenile idiopathic arthritis, and Sjogren’s syndrome [53,54]. The chronic use of CQ for the treatment of malaria and HCQ for rheumatological disorders has facilitated FDA approval for repurposing their use in cancer therapy.

The rationale behind the use of CQ and HCQ is to inhibit autophagy by deacidifying the lysosome and blocking the fusion of autophagosomes with lysosomes, thus preventing cargo degradation and the activation of programmed cell death. CQ and HCQ are also able to sensitize cancer cells to other chemotherapeutic agents, and are therefore used alone or in combination with systemic chemotherapy or radiotherapy [55].

Several concluded phase I and II clinical trials using CQ or HCQ alone or in combination with a chemotherapeutic agent in solid and non-solid tumors produced differing and not always encouraging results. Clinical trials investigating CQ and HCQ in several cancer types are reported in Table 2.

Of note, a phase I trial of HCQ in combination with the HDAC inhibitor vorinostat (VOR; 600 mg daily HCQ and 400 mg daily VOR) was performed in 19 patients with CRC. The combination of HCQ and VOR gave 2.9 months of median progression-free survival (PFS) in patients refractory to standard chemotherapy. Five of these 19 patients achieved prolonged stable disease. Adverse events were mild, and included fatigue and gastrointestinal disturbances. A phase II study was subsequently launched (NCT02316340) [56].

In another phase I clinical trial, 40 patients with melanoma were treated with HCQ and the alkylating agent temozolomide (TMZ). Patients received 200–1200 mg oral HCQ daily with 150 mg/m^2^ of dose-intense, oral TMZ. The combination was well tolerated, without any serious adverse effects (fatigue, anorexia, nausea, constipation, and diarrhea were reported). The highest dose level of HCQ (1200 mg daily) correlated with clinical benefit in nine out of 22 patients with refractory B-Raf Proto-Oncogene, Serine/Threonine Kinase (BRAF) wild-type melanoma [57].

Twenty-five patients with relapsed and refractory multiple myeloma were treated with HCQ plus bortezomib (BOR), which is a proteasome inhibitor widely used against myeloma. No toxicity was observed and, importantly, no exacerbation of neurotoxicity was reported. The recommended HCQ dose was determined to be 600 mg twice daily in addition to the standard dose of bortezomib. Since the only response was from patients who were BOR-naïve, the signal of activity was not considered sufficient to proceed to a phase II study [58].

A recent meta-analysis evaluating the efficacy of CQ and HCQ alone or in combination with additional treatments (chemotherapeutics/radiotherapy) in 293 patients affected by solid and non-solid cancers showed that both CQ and HCQ can greatly improve one-year overall survival, six-month PFS, and overall response rate (ORR), although to different degrees. The combination of CQ or HCQ plus gemcitabine yielded the best ORR in pancreatic adenocarcinoma, whereas the combination of CQ or HCQ plus TMZ and radiation gave the best six-month PFS in glioblastoma. The best ORR overall was obtained in non-Hodgkin lymphoma. In glioblastoma patients, autophagy inhibitors achieved the best six-month PFS and one-year overall survival, revealing new therapeutic opportunities to be further investigated. No significant improvement in ORR and six-month PFS was observed in patients with NSCLC or breast cancer [59].

Although autophagy is found hyperactivated in many tumor cells, only some cancer cell types are more sensitive to autophagy inhibitors, and this should inform clinical decisions. In addition, the timing of therapy is an important criterion to evaluate when constructing a treatment regimen.

Pancreatic ductal adenocarcinoma (PDAC) is one of the most aggressive and lethal cancers, with increasing incidence and few therapeutic options. The mutational activation of *KRAS* is the critical genetic driver of PDAC initiation and progression, and is essential for the maintenance of PDAC growth [60]. In addition, chronic activation of the RAS pathway is accompanied by high levels of autophagy. HCQ yielded a good response when used in combination with gemcitabine and paclitaxel in PDAC [61]. A recent study demonstrated that inhibition of the extracellular signal-regulated kinases (ERK) pathway in PDAC cells resulted in a massive activation of autophagic flux, theoretically making the cells more sensitive to autophagy inhibitors. PDAC cells treated with ERK inhibitors plus HCQ exhibited a high percentage of apoptosis. Extending these findings in vivo, the combination of ERK inhibitors plus HCQ in two heterogeneous patient-derived xenograft (PDX) models of *KRAS*-mutant pancreatic cancer showed that this combination blocked tumor progression and significantly extended survival [62].

Several studies also demonstrated that the phosphoinositide 3-kinase-protein kinase B/Akt (PI3K-PKB/Akt) modulators can have an effect on autophagy: class I PI3Ks are autophagy inhibitors, class II do have any specific effect on autophagy, and class III are autophagy activators, exerting their effect in the early step of autophagosome formation [63].

Methyladenine (3-MA) was the first PI3K inhibitor found to have a specific function on autophagic/lysosomal protein degradation. Recent and surprising findings have shown a dual role for 3-MA. The enzyme inhibits autophagy in low-nutrient cells, but in normal/high-nutrient conditions, it functions as class III PI3Ks, leading to autophagy induction by disrupting mTOR activity [64].

LY294002, another PI3K inhibitor, showed dose-dependent cell growth inhibition activity. However, the worsening of side effects prevented its entry into clinical trials [65].

Wortmannin, a fungal metabolite isolated in 1957, was originally used for its anti-inflammatory activity and later as a PI3K inhibitor. Wortmannin in combination with paclitaxel was reported to increase apoptosis and decrease tumor growth in a human ovarian cancer model [66]. 

Bafilomycin A1, a lysosomal H+ATPase inhibitor, inhibits lysosomal fusion to autophagosomes through binding between Beclin-1 and Bcl-2, although its effect is only short term [67].

### 5.2. Autophagy Activators

mTOR signaling is the second most frequently altered pathway in human cancers, and most drugs are able to induce autophagy by blocking the mTOR pathway (Table 3). Rapamycin was the first mTOR inhibitor identified using a target-based screening approach to induce cell cycle arrest [68]. Rapamycin binds FKBP12, which in turn releases RAPTOR from mTORC1 and RICTOR from mTORC2 [68]. The downstream effect of rapamycin leads to the inhibition of protein synthesis and cell cycle progression through the inactivation of 4EBP1 and S6K1, via the negative feedback regulation of hypoxia/VEGF signaling, blocking cancer progression [68].

mTOR inhibitors are classified into three generations:

1. Rapamycin and its analogs, known as rapalogs (everolimus or RAD001, temsirolimus or CCI-79, sirolimus, and ridaforolimus).

2. ATP-competitive inhibitors of mTORC1 and mTORC2, as well as PI3K/mTOR inhibitors (MLN0128, AZD2014, AZD8055, BEZ235, and CC223).

3. “Hybrid” molecules targeting mTORC that avoid resistance to first-generation and second-generation mTOR inhibitors, but not commonly used.

Rapalogs act with different mechanisms of action. They were initially used as immunosuppressants to avoid organ rejection. As they showed antiproliferative and anti-angiogenic effects, the potential use of these compounds in cancer treatment is being investigated [69]. Despite initially promising results, enthusiasm waned when several clinical trials showed that these inhibitors reduced cell proliferation without inducing cell death, suggesting that they are cytostatic and not cytotoxic. In addition, autophagy induction through mTORC1 inhibition may not be fully effective when mutations in mTOR signaling genes occur, preserving mTOR activity [69].

Metformin, commonly used for type II diabetes, was observed to reduce cancer incidence and stroke in diabetic patients. Metformin acts as a pro-autophagic drug, activating AMPK through the phosphorylation of LKB1 [70]. In HiMyc mice, metformin administrated at 250 mg/kg led to autophagy activation and the repressed proliferation of prostate cancer (PCa) cells [71]. In addition, at 10 nM, it suppressed the growth of glioblastoma cells and decreased mitochondria-dependent ATP generation in combination with diclofenac [72]. Preclinical studies showed a synergistic effect of chemotherapy and radiotherapy in combination with metformin, which has not yet been approved [73].

Perifosine induces autophagy through AKT inhibition, which interferes with PI3K signaling, thus affecting apoptosis, proliferation, and inflammation. Perifosine is currently in Phase III clinical trials for the treatment of CRC in combination with capecitabine [74]. In clinical trials, it is also used in multiple myeloma patients in combination with bortezomib and dexamethasone [75].

## 6. Natural Compounds as Inhibitors or Activators of Autophagy

Several natural compounds derived from plants, marine organisms, and microorganisms have attracted considerable interest for their ability to inhibit cancer cell proliferation and target altered pathways. To date, many natural molecules have been shown to inhibit or activate autophagy, paving the way toward a new era of drug discovery and the development of novel compounds with potential anti-cancer activity.

Artemisinin, isolated from the medicinal herb *Artemisia annua*, is widely used in malaria treatment and is able to kill cancer cells via autophagy modulation [76]. In lung cancer cells, artemisinin inhibited cell proliferation and activated apoptosis, exerting a stronger effect in combination with CQ [77]. A metabolite of artemisinin, dihydroartemisinin (DHA), suppressed NF-kB activation [78] and the accumulation of ROS [77], and induced autophagy in different cancer cell lines [77,78]. By inhibiting mTOR, DHA induced autophagy and apoptosis in cisplatin-resistant ovarian cancer cells [79].

Curcumin is a polyphenolic compound derived from turmeric that is used to treat a variety of disorders such as asthma, anorexia, allergies, sinusitis, and bronchial hyperactivity. Curcumin was also shown to play a hepatoprotective, cardioprotective, chemopreventive, anti-inflammatory, and anticarcinogenic role [80]. It acts on different molecular targets including NF-kB, AKT [81], and signal transducer and activator transcription (STAT3) [82], and induces apoptosis in cancer cells in vitro and in vivo. Curcumin-induced autophagy was shown to be mediated by ROS generation, Beclin-1 upregulation, and consequently LC3-II accumulation, leading to cell death in CRC cells [83]. When tested in vitro, curcumin induced autophagy in mesothelioma and K562 chronic myelogenous leukemia cells via modulation of the PI3K/AKT/mTOR and NF-kB signaling pathways [84]. In addition, in vivo studies in a PCa cell xenograft mouse model showed that the administration of curcumin increased median survival, reducing tumor size [85]. Curcumin enhanced autophagic flux in HCT116 human colon cancer cells by activating TFEB [86].

Resveratrol, which belongs to the group of polyphenolic stilbenoids, is present in grapes, berries, and other plant sources [87]. Resveratrol has anti-inflammatory, anti-cancer, antiangiogenic, and anti-invasion effects, and is able to negatively modulate cell cycle progression [87]. A link between autophagy and resveratrol is reported in several cancers. For example, resveratrol is able to induce autophagy-mediated cell death in ovarian cancer, and inhibit breast cancer stem cell proliferation by inducing autophagy and suppression of the Wnt/β-catenin pathway [88].

Many studies underscore the importance of resveratrol in autophagy activation. In U251 human glioma cells, resveratrol delayed apoptosis through autophagy induction, and in cervical cancer cells, it inhibited the activation of NF-kB, leading to an increase in lysosomal permeability, suggesting autophagy-mediated mechanisms of cell death [89]. In MCF-7 breast cancer cells, which lack caspase-3 activity, resveratrol stimulated noncanonical autophagy, providing a novel mechanism of cell death in these cells [90]. Additionally, in imatinib-resistant chronic myelogenous leukemia cells, resveratrol-mediated cell death occurs through autophagy activation via the JNK-dependent accumulation of p62 [91].

## 7. mTOR Inhibitors in Clinical Trials

The following subsections focus on the use of mTOR inhibitors in human clinical trials in different cancer types (Table 4).

### 7.1. Breast Cancer

Breast cancer is the most common form of cancer in women, and although the survival rate is increasing, it is still the first cause of death among the female population. The PI3K/AKT pathway is hyperactivated in breast cancers due to frequent mutations in the *PI3K* gene [92]. The Cancer Genome Atlas reports *PI3K* mutations in 35% of hormone receptor (HR)-positive cancers, 23% of human epidermal growth factor receptor 2-positive cancers, and almost 10% of triple negative breast cancers [92]. Everolimus was the first rapalog approved for the treatment of metastatic HR-positive breast cancers in combination with endocrine therapy, which is the gold standard for these neoplasms [92]. The completed BOLERO-2 trial showed that combination therapy with everolimus plus the aromatase inhibitor exemestane doubled PFS (from 4.1 months to 10.6 months) and displayed a manageable safety profile. Overall survival improved to 31 months compared to 26 months with hormone therapy alone [93,94]. These results led to the FDA approval of everolimus plus exemestane in advanced metastatic breast cancers. Everolimus is now under investigation in combination with other drugs followed by a better patient stratification [95]. In addition to rapalogs, promising results were obtained with ATP competitors such as MLN0128, AZD2014, AZD8055, and CC223 in breast cancer cell lines and xenograft experiments, moving them toward approval for phase II clinical trials [92,96,97].

### 7.2. Ovarian Cancer

Ovarian cancer is the second most lethal gynecologic cancer [98]. Although most patients initially respond well to cytoreduction surgery and chemotherapy, many eventually develop recurrent disease and resistance to chemotherapy. Preclinical studies on ovarian tissue samples of ovarian cancer patients showed that the mTOR pathway is upregulated, indicating that rapalogs may be a useful therapeutic option [99]. A phase II clinical trial investigating the combination of everolimus plus the aromatase inhibitor letrozole in a small number of patients with relapsed estrogen receptor-positive high-grade ovarian cancer showed that this treatment is associated with a promising 12-week PFS [100]. Chemoresistance is the major challenge in current ovarian cancer therapy, and the mechanism is not completely understood. Several findings showed that chemoresistance is associated with epithelial–mesenchymal transition (EMT) and cancer stem cell (CSC) marker expression in ovaries [101]. A recent study reported that EMT and CSC marker expression were significantly increased in chemoresistant ovarian cancer cells, accompanied by the activation of PI3K/AKT/mTOR signaling [102]. Treatment with BEZ235, a dual PI3K/mTOR inhibitor, and the well-known drug cisplatin significantly inhibited the PI3K/AKT/mTOR signaling pathway, reversed EMT, and decreased CSC marker expression in chemoresistant ovarian cells compared to monotreatment with cisplatin [101]. Therefore, combination therapy with cisplatin and BEZ235 may be a promising option to treat ovarian cancer patients.

### 7.3. Prostate Cancer

PCa is the second most prevalent cancer in men worldwide, and despite efforts to understand the underlying mechanisms driving PCa development and progression, it is the fifth most common cause of cancer death in men [103,104].

Genetic alterations of the mTOR pathway were found in 42% of primary prostate tumors [105], and the constitutive activation of several members of this pathway was observed in PCa cell lines, xenograft models, and 30–50% of primary tissues. Although preclinical studies yielded promising results, the first clinical trials failed. The last published study appeared in 2018, in which the dual mTOR inhibitor MLN0128 was developed to assess antitumor activity in metastatic castration-resistant PCa [106]. The clinical efficacy of MLN0128 in this form of PCa was limited due to drug toxicity, an increase in PSA levels suggesting hormone reactivation resulting from mTOR inhibition, and the poor inhibition of mTOR signaling targets. In preclinical studies, encouraging results were obtained with BEZ235 combined with the androgen receptor antagonist enzalutamide, which induced efficient tumor reduction in a PCa mouse model [107]. BEZ235 is currently under investigation in phase I/II clinical trials alone and in combination in castration-resistant PCa patients.

### 7.4. Thyroid Cancer

In the USA, thyroid cancer is the fifth most common cancer in women, and its incidence continues to rise worldwide [108]. Differentiated thyroid cancer is the most frequent subtype and standard treatment; thyroidectomy followed by either radioactive iodine or observation is effective [109]. However, there are still no successful treatments for patients with medullary and anaplastic thyroid cancers, two rare but aggressive forms of the disease. Some promising results were obtained with anti-angiogenesis inhibitors targeting multiple receptor tyrosine kinases (including vandetanib, lenvatinib, and cabonzatinib), but with serious adverse effects [110,111] (NCT02657369, NCT02244463, NCT01240590, NCT02239900).

Although these treatments result in prolonged PFS, they are not curative and are used in patients with progressive or symptomatic disease [108]. A multicenter phase II study assessed the efficacy and safety of everolimus in locally advanced or metastatic thyroid cancer of any histology. Thyroid cancer patients resistant to standard therapy received everolimus [112] (10 mg/daily orally) until high toxicity or disease progression. Of the 40 enrolled patients, disease control rate and objective response was achieved in 81% and 5% of patients, respectively. Treatment yielded stable disease (76%) and progressive disease (17%) among patients; the median PFS was 47 weeks. Mucositis, anorexia, and aspartate transaminase/alanine transaminase were the most common adverse events [113].

Anaplastic thyroid cancer is the highest aggressive form of the disease, and represents more than 50% of all thyroid cancer deaths. mTOR inhibition was found to exert antitumor activity in this cancer subtype in preclinical studies. A phase I/II clinical trial evaluating everolimus in a very small cohort of patients showed a partial response in one patient (27.9 months) and stable disease in two patients (3.7 and 5.9 months). Genomic analysis detected mutations in the PI3K/mTOR pathway in these latter two patients [113], highlighting that mTOR is a druggable pathway in this cancer.

### 7.5. Gastrointestinal Cancer

The PI3K/AKT pathway was found disrupted in 60% of gastric cancers and 15% of CRCs. *PTEN*, an inhibiting PI3K pathway, is a tumor suppressor gene that is frequently altered in these cancer types [63,114]. Preclinical studies performed in gastric cell lines showed a significant suppression of cell proliferation by everolimus and sirolimus [115]. Although rapamycin yielded good results in gastric cell lines as well as in a xenograft mouse model, only everolimus was tested in a phase III trial in patients with advanced gastrointestinal cancer. The results of the trial demonstrated acceptable side effects and a PFS of six months (NCT00879333) [116].

Immunohistochemical analysis revealed the consistent activation of mTORC1 in colorectal adenoma and CRC, indicating a direct effect of mTORC1 activation on the initiation and progression of this cancer. However, the effects of everolimus or temsirolimus were limited in CRC patients, and these drugs were therefore abandoned as monotherapy. Combination therapies with rapalogs and VEGF inhibitors, the somatostatin analog octreotide, or MEK inhibitors are under investigation, showing encouraging results [117].

### 7.6. Lung Cancer

AKT phosphorylation and the consequent upregulation of the PI3K pathway is present in 50–70% of patients with NSCLC [118]. Inhibitors of this pathway such as everolimus, temsirolimus, and ridaforolimus were used to treat NSCLC patients in clinical trials, but failed to show any promising results. Everolimus and ridaforolimus yielded encouraging results in a phase I study, but did not progress to phase II trials in NCLSC due to their toxicity. Temsirolimus is known to suppress cell proliferation in NSCLC [119]. In a phase I clinical trial, only one out of 63 patients had a good response to treatment with temsirolimus. However, in a phase II clinical trial evaluating the drug at different doses, 35% of patients achieved a good response, while 27% had stable disease. Treatment with both everlimus and temsirolimus caused several adverse effects, such as nausea, fatigue, stomatitis, dyspnea, mucositis, and asthenia [119]. The combination of everolimus plus chemotherapy or radiotherapy also failed to show any positive results in NSCLC patients [120,121]. In a phase II study in advanced NSCLC, patients treated with chemotherapy or chemotherapy and epithelial growth factor receptor inhibitors, combined with everolimus (10 mg/day) reached a response rate of 4.7% and a disease control rate of 47.1% [122,123]. Everolimus showed a possible inhibition of NSCLC cells in a preclinical study [124]. Phase I and phase II clinical trials with sirolimus and other therapies in patients with NSCLC are still ongoing [124].

### 7.7. Renal Cell Carcinoma

Renal cancer is classified as either clear cell RCC (ccRCC, 85%), papillary RCC (0–15%), chromophobe RCC (5%), or collecting duct carcinoma and medullary carcinoma (1%) [125]. The mTOR pathway is involved in regulating cell metabolism, and RCC is associated with metabolic dysregulation [63]. Temsirolimus and everolimus are able to inhibit mTORC1 activation, leading to survival benefits in advanced ccRCC patients (Global ARCC trial) [126,127]. In another study, patients carrying *mTOR*, *TSC1,* or *TSC2* mutations [128] treated with mTOR inhibitors (everolimus and temsirolimus) were found to benefit more patients with rapid disease progression [129].

### 7.8. Leukemia

Acute lymphoblastic leukemia (ALL) is a disease of lymphoid progenitor cells that occurs in both pediatric and adult patients; 75% of ALLs develop from precursors of B-cell lineage, and the remainder develops from T-cell precursors. Genomic analysis revealed a huge number of recurrent mutations in different pathways such as NOTCH, JAK-STAT, MAPK, and PI3K/AKT/mTOR [130]. Targeting PI3K/AKT/mTOR pathway signaling was investigated in different preclinical models of ALL, showing the efficacy of mTOR drugs used in association with chemotherapy [131]. Since standard treatments (chemotherapy and radiotherapy) are not curative and relapse of the malignancy occurs, new approaches using mTOR inhibitors are being developed [131].

In clinical trials of ALL, rapamycin interferes with cytokine signaling, through interaction with its intracellular receptor (FK506-binding protein 12) [131]. As the pharmacological effect of rapamycin has certain limitations, a number of derivatives were developed with less immunosuppressive activity and increased antitumor action [132]. Both in vivo and in vitro model studies of the well-known rapalog RAD001 reported good results in terms of its antiproliferative activities. RAD001 is able to induce caspase-independent cell death and cell cycle regulation changes, and may be effective in overcoming resistance in ALL [131].

T-cell ALL (T-ALL) is caused by different genomic lesions affecting the development of T-cells; high mTOR expression levels are reported to be more frequent in adults than in children [133]. Rapamycin and its derivatives could be used for the treatment of T-ALL in combination with other drugs, such as doxorubicin [134], cyclophosphamide, and methotrexate [135] (NCT00968253, NCT01184885). Recent studies showed a good synergistic effect in the combined use of the CDK4/6 inhibitor LEE-01 (ribociclib) with RAD001 and glucocorticoids in the treatment of T-ALL and B-cell ALL (B-ALL) (NCT03328104) [134].

The PI3K/AKT/mTOR pathway can be activated by NOTCH1 during T-cell development [132]. The inhibition of NOTCH1 is associated with mTOR suppression, thus highlighting the crosstalk between the two signaling cascades, and different PI3K upstream signaling receptors are upregulated by NOTCH1 in T-cell progenitors [136]. The PI3K/mTOR inhibitor PKI-587 (gedatolisib) displayed an inhibitory effect on T-ALL cells and efficacy in T-ALL patients with poor prognosis. In T-ALL cells, PKI-587 was able to block proliferation and colony formation, and delayed tumor progression in immune-deficient mouse models. In CRLF2/JAK-mutant models, treatment with PKI-587 induced a 92.2% reduction in leukemia cell proliferation [137]. Preclinical studies of the PI3K/mTOR inhibitor BEZ235 showed antiproliferative effects in ALL cell lines by inducing antileukemic activity when associated with glucocorticoids in in vitro and in vivo models [138].

The administration of RAD001 in combination with CC1-779, hyperfractionated cyclophosphamide, vincristine, doxorubicin, and dexamethasone (hyper-CVAD) chemotherapy in T-ALL or B-ALL patients inhibited the phosphorylation of S6RP, which is a downstream target of mTOR. Interestingly, the combination of RAD001 and hyper-CVAD did not induce toxicity compared to treatment with hyper-CVAD alone (NCT00968253) [139]. A toxicity study is investigating the use of CVAD in combination with dexamethasone, cyclophosphamide, and etoposide in children with relapsed ALL (NCT01614197) [140]. Rapamycin in combination with chemotherapy with or without donor stem cell transplant in adult patients with Philadelphia chromosome-positive ALL is under investigation (NCT00792948) [140].

Although the combination of chemotherapy drugs with targeted mTOR inhibitors seems to be tolerated with low toxicity in ALL treatment, further studies will be necessary to define the most effective pharmacological protocols and drug doses to reduce adverse effects.

## 8. Conclusions and Future Directions

A large body of evidence shows that autophagy is activated in tumorigenesis, and modulating this process may therefore be effective in tackling cancer [141]. Although this idea is generally accepted, autophagy-targeting drugs have yielded contradictory or limited results due to the lack of a precise understanding of their molecular mechanisms.

Taking account of available results, a number of shortcomings in autophagy modulation should be addressed in order to improve therapeutic strategies.

First, tumor heterogeneity: the existence of subpopulations of cells with distinct genotypes and phenotypes might contribute to inducing different autophagic levels within a primary tumor and its metastases, or between tumors of the same histopathological subtype.

Second, the quantification of autophagy: although many tools have been developed to measure autophagic flux, these methods are not totally reliable in vivo, due to compensatory mechanisms activated by tumor cells and for adverse side effects in patients. Chemical studies will help to identify inhibitors achieving high efficacy and low toxicity in cancer patients.

Third, the timing of drug administration: robust and rapid inhibition of autophagy flux in cancer cells initially induces apoptosis as well as potentiates adjuvant therapies (i.e., chemotherapy and/or radiotherapy). Conversely, late autophagic inhibition enables cancer cells to escape cell death by activating a pro-survival program.

Therefore, taking together these observations, performing tissue-based autophagy assay immediately after surgery will likely reveal whether the pathway is altered or not in the tissue of interest, allowing patient-tailored cancer treatments.

Furthermore, massive next-generation sequencing analysis will elucidate the molecular aspects of aberrant signal transduction activated in different cancer types and identify the most suitable druggable molecules or gene mutations, allowing the development of more precision therapies. The stratification of patients remains the fundamental step to identifying the most effective therapeutic approach in different cancers.

## Figures and Tables

**Figure 1 cancers-11-01465-f001:**
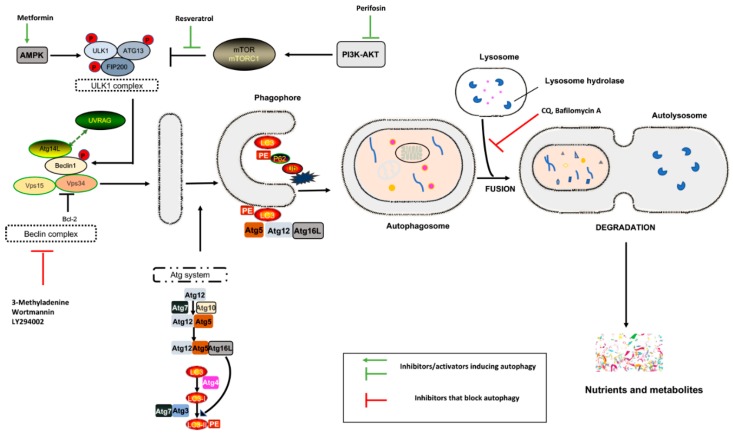
Schematic representation of regulatory pathways involved in the autophagic event and point of inhibition/activation. Under a deprivation of nutrients or growth factors, ULK activation occurs via the activation of activating AMP kinase (AMPK) and/or the inhibition of mTOR. ULK functions in a complex with FIP200 and Atg13, which phosphorylates Beclin-1, leading to VPS34 activation and phagophore formation. Association between Beclin-1 and Bcl-2 inhibits autophagy. Two ubiquitin-like conjugation systems are engaged, one involving Atg12, Atg5, and Atg16L proteins, and the other converting LC3 protein from its LC3I form to LC3II. This event leads to closure of an elongated phagophore with the formation of a mature autophagosome, which is followed by transport of the autophagic cargo to lysosomes, degradation of this cargo by lysosomal hydrolases, and recycling of the products for use in metabolism.

**Figure 2 cancers-11-01465-f002:**
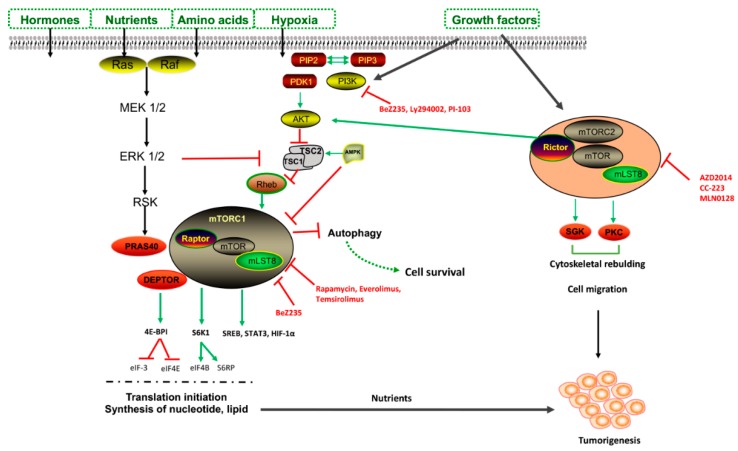
Schematic representation of mechanistic target of rapamycin complex 1 (mTORC1) and mTORC2 pathways and point of inhibition. mTORC1 associates with endosomal and lysosomal membranes via its effector, mTORC2. Once phosphorylated, AKT can activate mTORC1 directly, either by phosphorylating and dissociating the proline-rich Akt substrate of 40kDa (PRAS40) from the regulatory-associated protein of mTOR (RAPTOR) or by inhibiting tuberous sclerosis (TSC)1/2 complex formation, releasing the Ras homolog enriched in brain (RHEB), which is an activator of mTORC1. Protein translation and the synthesis of nucleotide lipid via 4E-BP1 and S6K1 is regulated by mTORC1. In tumorigenesis, mTORC2 activates signal transducer and activator transcription (STAT3), hypoxia-inducible factor 1a (HIF1a), and protein phosphatase 2A (PP2A). In addition, mTORC2 regulates serum glucose kinase (SGK) and protein kinase (PKC) to induce cell survival, cytoskeleton organization, and cell migration. mTORC1 functions as a negative regulator of autophagy, exerting its inhibitory action by phosphorylating and inactivating ULK1/2 and Atg13.

**Table 1 cancers-11-01465-t001:** Autophagy inhibitors.

Name	Mode of Action	Structure
Choloroquine	Endosomal acidification inhibitor	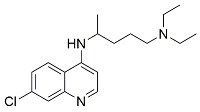
3-Methyladenine	PI3K inhibitor	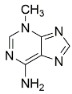
Bafilomycin A1	Endosomal acidification inhibitor	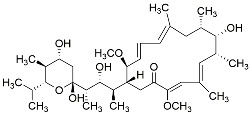
LY294002	PI3K inhibitor	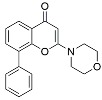
SB202190	MAPK inhibitor	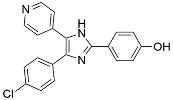
SB203580	MAPK inhibitor	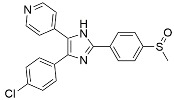

**Table 2 cancers-11-01465-t002:** Clinical trials involving hydroxychloroquine and chloroquine. CQ: chloroquine, HCQ: hydroxychloroquine, TMZ: temozolomide, VOR: vorinostat.

Disease	Trial	Autophagy Inhibitor	No. Patients	Adverse Effects	Dose	Ref.
Colorectal cancer	Phase I, vorinostat + HCQ	HCQ	19	Fatigue and gastrointestinal disturbances	VOR 600 mg/daily, HCQ 400 mg/daily	[56]
Melanoma	Phase I, HCQ + TMZ	HCQ	40	Fatigue, anorexia, nausea, constipation, and diarrhea	HCQ 200-1200 mg/ daily + TMZ 150 mg/m2	[57]
Refractory multiple myeloma	Phase I, HCQ + BOR	HCQ	25	None	HCQ 600 mg/daily + standard dose of BOR	[58]
Glioblastoma	Phase III, CQ + TMZ + radiation	CQ	30	None	150 mg/daily	[59]
Non-small cell lung cancer	Phase II, CQ + whole-brain radiation	CQ	73	None	150 mg/daily	[59]
Glioma	Randomized, double-blind phase II, carmustine, radiation, and chloroquine	HCQ	30	None	HCQ 150 mg/daily	[59]
Breast Cancer	Phase II, everolimus + HCQ	HCQ	60	None	HCQ 150 mg/daily	[59]

**Table 3 cancers-11-01465-t003:** Autophagy activators. mTOR: mechanistic target of rapamycin complex, NF-kB: nuclear factor kappa-light-chain-enhancer of activated B cells.

Name	Mode of Action	Structure
Rapamycin	mTOR inhibition–TLR signaling	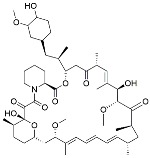
Everolimus	mTOR inhibition	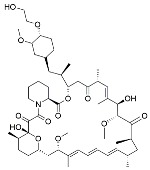
Metformin	AMPK activation	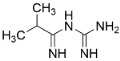
Perifosine	AKT inhibition	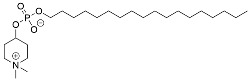
Resveratrol	NF-kB inhibition	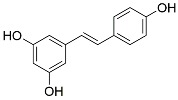

**Table 4 cancers-11-01465-t004:** mTOR inhibitors in clinical trials. B-ALL: B-cell ALL, NSCLC: non-small cell lung cancer, T-ALL: T-cell ALL.

Disease	Autophagy Effect	Drug(s)	No. Patients	Refs
Breast cancer (HR^+^)	mTORC1 inhibition mTORC1 and mTORC2 inhibition	Everolimus + aromatase inhibitor exemestane. MLN0128 AZD2014 CC-223	724 Experiments in cell lines and xenograft models	[94] [96,97]
Ovarian cancer	mTORC1 inhibition PI3K/mTOR inhibition	Everolimus + aromatase inhibitor letrozole BEZ235 + cisplatin	20 Ovarian cancer stem cells	[100] [101]
Prostate cancer	mTORC1 inhibition	MLN0128 BEZ235	9 Prostate cancer cells	[106] [107]
Thyroid cancer Anaplastic thyroid cancer	mTORC1 inhibition	Everolimus Everolimus	40 2	[112] [113]
Gastrointestinal cancers	mTORC1 inhibition	Everolimus	656	[116]
Lung cancer (NSCLC)	mTORC1 inhibition	Everolimus Everolimus Chemoterapy + EGFr inhibition + everolimus Temsirolimus	92 26 85 63	[120] [120] [122] [57]
Renal cell carcinoma	mTORC1 inhibition	Everolimus (RAD001) Rapamycin + doxorubicin	41	[127]
Leukemia T-ALL	mTORC1 inhibition	Rapamycin + cyclophosphamide Rapamycin + methotrexate	7 7	[134] [135]
T-ALL/B-ALL	mTORC1 inhibition	RAD001 (everolimus) + LEE-01 + glucocorticoids	15	[134]
T-ALL	PI3K/mTOR inhibition	PKI-587 (Gedatolisib) BEZ235 + cytarabine (AraC) or doxorubicin or dexamethasone		[137] [138]
B-ALL/T-ALL	mTORC1 inhibition	Everolimus (RAD001) + vincristine + doxorubicin + cyclophosphamide + dexamethasone	22	[139]
ALL + Philadelphia chromosome-positive ALL	mTORC1 inhibition	Rapamycin + chemotherapy +/− stem cell transplant in patients	97	[140]

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
