# Peer review of "Autophagy Function and Dysfunction: Potential Drugs as Anti-Cancer Therapy"

_cancers, 2019, doi:10.3390/cancers11101465_

Round 1
Reviewer 1 Report
In general, the text is very long, and too descriptive; needs to be rewritten in scientific form not a story telling form.
Many unnecessary indents and new paragraphs, even though the idea is the same. Also many typos and missing punctuations.
Many poorly cited work, which could fall into plagiarism. Example: last paragraph in page 2 explaining the phagophore engulfment without citing any papers.
The "Autophagy and Signalling Pathways" seem to be oriented around mTOR pathways.. many other pathways are forgotten such as one of the major regulators of autophagy: AMPK. Also several downstream targets of mTOR that are now recognized as the master transcriptional regulators of autophagy and lysosomal biogenesis: MITF, TFEB, and TFE3 are not even mentioned. The general idea I got about this review is that it is outdated and seem to be a mere repetition of what has already been shown before by other reviews on the same topic.
The "Role of Autophagy in Cancer" part was good and thorough; even though-again-too descriptive- and goes into unnecessary details, which shifts the reader from the main idea of the review. The part about mTOR inhibitors however was not so straight forward, as mTOR inhibitors are usually used to block protein synthesis and halt proliferation, their effect on autophagy wasn't clearly studied and hence this needs to be mentioned in the text so the idea is clear; especially that the point of this review is autophagy in general not mTOR (which really seems to be the center of this review).
Author Response
In general, the text is very long, and too descriptive; needs to be rewritten in scientific form not a story telling form.
Thank you for the comment. We extensively modified the text, eliminating redundancy and ‘story telling’ form
Many unnecessary indents and new paragraphs, even though the idea is the same. Also many typos and missing punctuations.
We revised the manuscript, eliminating unnecessary paragraphs and indents.
Many poorly cited work, which could fall into plagiarism. Example: last paragraph in page 2 explaining the phagophore engulfment without citing any papers.
We apologize for this, adding the missed references (2,3).
The "Autophagy and Signalling Pathways" seem to be oriented around mTOR pathways.. many other pathways are forgotten such as one of the major regulators of autophagy: AMPK. Also several downstream targets of mTOR that are now recognized as the master transcriptional regulators of autophagy and lysosomal biogenesis: MITF, TFEB, and TFE3 are not even mentioned. The general idea I got about this review is that it is outdated and seem to be a mere repetition of what has already been shown before by other reviews on the same topic.
Thank you for the comment. We modified the paragraph ‘Autophagy and Regulatory Machinery’, adding the other regulatory pathways than mTOR. Accordingly, we modified also the paragraph 'Autophagy and Cancer' including the activation of other regulators in cancer.
The "Role of Autophagy in Cancer" part was good and thorough; even though-again-too descriptive- and goes into unnecessary details, which shifts the reader from the main idea of the review. The part about mTOR inhibitors however was not so straight forward, as mTOR inhibitors are usually used to block protein synthesis and halt proliferation, their effect on autophagy wasn't clearly studied and hence this needs to be mentioned in the text so the idea is clear; especially that the point of this review is autophagy in general not mTOR (which really seems to be the center of this review).
According to reviewer’s suggestion, we extensively modified the text of the paragraph, trying to better clarify the point.
Reviewer 2 Report
In this review, the authors described the role of autophagy in cancer, and the potential drugs via regulation of autophagy activity as anti-cancer therapy. This was accompanied by a summary of autophagy’s functions in cancer cells and explanation of potential anticancer drugs that modulate autophagy for therapeutic targeting in cancer. There are major corrections and suggestions to improve the quality of the manuscript and these are listed as follows:
The authors should describe the relationship between autophagy and apoptosis in detail regarding cancer.
Autophagy and Regulatory Machinery Section
There is controversy on the role of P62 on autophagy among various studies, specifically regarding whether it is upregulated or downregulated in the autophagy pathway. The authors might need to note on this controversy and describe their stance on changes in levels of P62 with respect to autophagy.
Figure 2
It is difficult to understand the schema in Figure 2. The authors might need to better represent the signaling pathway of the study. The authors might also need to combine the figures 1 and 2 to represent the holistic picture of autophagy.
Autophagy and Signaling Pathways
The discussion is mainly about mTOR, but since the title of the subsection is “autophagy and signaling pathways”, the authors need to include other autophagy pathways.
The authors might need to add a reaction mechanism linking mTOR and autophagy under stress condition.
Figure 3
What Figure 3 tries to elaborate on is vague regarding the relationships between mTOR and autophagy and between autophagy and tumorigenesis. The authors might need to more clearly describe the interactions between mTOR and autophagy and autophagy and tumorigenesis.
Role of Autophagy in Cancer
The manuscript below is not completely inclusive of the term “double-edged sword” since you only focus on the elevated activity of autophagy, not including the attenuated activity. The authors might need to include the effect of downregulated autophagy in cancer to describe the effects of bilateral activity levels of autophagy and their effects on cancer suppression or induction.
The manuscript about activation of Erk via B-RAF needs another reference paper (Reference Paper 26) that describes the proposed activation. The reference you gave does not describe the above pathway.
Should we try to enhance or to inhibit autophagy in cancer?
The authors should include other anti-cancer drugs that utilize autophagy related signaling pathways additionally other than mTOR pathway.
The authors should describe various types of anticancer drugs that specifically regulate autophagy and their underlying mechanisms.
The authors must communicate on the relationship between autophagy and mitochondrial respiration.
Minor Review
The authors need additional grammar editing on the manuscript.
The authors mentioned ‘prostate cancer mouse model p62’. However, the reference paper used a different terminology to describe the model. The authors should make the terminology consistent with the one used by the reference paper.
Artemisinin
The authors should include the reference about “In lung cancer cells—” in the first paragraph.
Author Response
In this review, the authors described the role of autophagy in cancer, and the potential drugs via regulation of autophagy activity as anti-cancer therapy. This was accompanied by a summary of autophagy’s functions in cancer cells and explanation of potential anticancer drugs that modulate autophagy for therapeutic targeting in cancer. There are major corrections and suggestions to improve the quality of the manuscript and these are listed as follows:
The authors should describe the relationship between autophagy and apoptosis in detail regarding cancer.
Thank you for your comment- According to reviewer’s suggestion, we added a new paragraph on the relationship between autophagy and apoptosis (Pg 7-8)
Autophagy and Regulatory Machinery Section
There is controversy on the role of P62 on autophagy among various studies, specifically regarding whether it is upregulated or downregulated in the autophagy pathway. The authors might need to note on this controversy and describe their stance on changes in levels of P62 with respect to autophagy.
We apologize for the lack of clarity on the role of p62. We modified the text to better clarify the point.
Figure 2
It is difficult to understand the schema in Figure 2. The authors might need to better represent the signaling pathway of the study. The authors might also need to combine the figures 1 and 2 to represent the holistic picture of autophagy.
Thank you for the suggestion. We combined the figure 1 and 2 in the new figure 1, adding also the point of activation/inhibition
Autophagy and Signaling Pathways
The discussion is mainly about mTOR, but since the title of the subsection is “autophagy and signaling pathways”, the authors need to include other autophagy pathways.
Thank you for the comment. We included the other autophagy pathways in the new paragraph ‘Autophagy and Regulatory Machinery’, eliminating the old ‘Autophagy and Signaling pathway’.
The authors might need to add a reaction mechanism linking mTOR and autophagy under stress condition.
According to this, we added a clearer description of autophagy activation under stress conditions (pg 4)
Figure 3
What Figure 3 tries to elaborate on is vague regarding the relationships between mTOR and autophagy and between autophagy and tumorigenesis. The authors might need to more clearly describe the interactions between mTOR and autophagy and autophagy and tumorigenesis.
The old figure 3 is now figure 2 , in which we clearly describe the interactions between mTOR pathway and tumorigenesis, drawing all the molecular components and the drug acting on the pathway activation.
Role of Autophagy in Cancer
The manuscript below is not completely inclusive of the term “double-edged sword” since you only focus on the elevated activity of autophagy, not including the attenuated activity. The authors might need to include the effect of downregulated autophagy in cancer to describe the effects of bilateral activity levels of autophagy and their effects on cancer suppression or induction.
According to the comment, we modified the text highlighting the differences in downregulated and upregulated autophagy pathway in cancer.
The manuscript about activation of Erk via B-RAF needs another reference paper (Reference Paper 26) that describes the proposed activation. The reference you gave does not describe the above pathway.
We apologize. We corrected and checked all references
Should we try to enhance or to inhibit autophagy in cancer?
The authors should include other anti-cancer drugs that utilize autophagy related signaling pathways additionally other than mTOR pathway.
Thank you for the comment. We added additional anti-cancer drugs in the paragraph ‘’Should we try to enhance or to inhibit autophagy in cancer?, as requested
The authors should describe various types of anticancer drugs that specifically regulate autophagy and their underlying mechanisms.
We tried to give a paramount picture of anticancer drugs that specifically modulate the autophagy and their mechanism of action, both in the text and in the figures and in the tables
The authors must communicate on the relationship between autophagy and mitochondrial respiration.
Thank you for the comment. We added in the paragraph ‘Autophagy and apoptosis’ the crosstalk between autophagy and mitochondrial respiration, highlighting the central role of mitochondria in cellular homeostasis
Minor Review
The authors need additional grammar editing on the manuscript.
We apologize and we edited and revised the manuscript with the help of a mother-tongue expert
The authors mentioned ‘prostate cancer mouse model p62’. However, the reference paper used a different terminology to describe the model. The authors should make the terminology consistent with the one used by the reference paper.
We corrected accordingly.
Artemisinin
The authors should include the reference about “In lung cancer cells—” in the first paragraph.
We corrected accordingly.
Reviewer 3 Report
This manuscript, “Autophagy function and dysfunction: potential drugs as anti-cancer therapy” by Cuomo et al., reviews literature on autophagy dysregulation in cancer and completed/ongoing drug trials on autophagy inhibitors/mTOR inhibitors (autophagy activators) in various cancers as cancer therapeutics. The authors discuss the dual role of autophagy in inhibition of cancer in early stages but promotion of cancer progression in late stages. Finally, the authors also review the literature on potential anticancer activity of natural compounds by autophagy inhibition/activation.
1. It will add clarity to this review manuscript if potential anti-cancer drugs can be clearly classified as autophagy inhibitors or autophagy activators (mTOR inhibitors).
2. Figure 3 can be modified to show the targets for various anticancer drugs discussed in this manuscript or a separate schematic figure can be added to show how these drugs alter autophagy in cancer.
3. The language of this review manuscript can be more interactive. In addition to the review of existing literature the authors can further elaborate the discussion to provide their viewpoint on repurposing, benefits and prospects of autophagy altering drugs in successful cancer treatment.
Author Response
This manuscript, “Autophagy function and dysfunction: potential drugs as anti-cancer therapy” by Cuomo et al., reviews literature on autophagy dysregulation in cancer and completed/ongoing drug trials on autophagy inhibitors/mTOR inhibitors (autophagy activators) in various cancers as cancer therapeutics. The authors discuss the dual role of autophagy in inhibition of cancer in early stages but promotion of cancer progression in late stages. Finally, the authors also review the literature on potential anticancer activity of natural compounds by autophagy inhibition/activation.
It will add clarity to this review manuscript if potential anti-cancer drugs can be clearly classified as autophagy inhibitors or autophagy activators (mTOR inhibitors).
Done. Thank you for notice this. we modified Table 1 and Table 3 accordingly and the text.
Figure 3 can be modified to show the targets for various anticancer drugs discussed in this manuscript or a separate schematic figure can be added to show how these drugs alter autophagy in cancer.
Thank you for the comment. We modified the old figure 3 accordingly that is now Figure 2 adding how drugs act on the pathway.
The language of this review manuscript can be more interactive. In addition to the review of existing literature the authors can further elaborate the discussion to provide their viewpoint on repurposing, benefits and prospects of autophagy altering drugs in successful cancer treatment.
We added in the paragraph Conclusion and future directions, our viewpoint on how the autophagy modulators could be successful in cancer treatment.
Round 2
Reviewer 1 Report
The manuscript is now in acceptable form.
Author Response
Thank you for the comments.
Reviewer 2 Report
In this review, the authors described the role of autophagy in cancer as well as the potential agents that exert anticancer activity by regulation of autophagy. This was accompanied by summary of autophagy’s functions in cancer cells and explanation of potential anticancer drugs that modulate autophagy for therapeutic targeting in cancer. There are minor corrections and suggestions to improve the quality of the manuscript, and these are listed as in the following:
1. Authors should improve the formatting of the manuscript
2. Authors should revise the references that use the more recently updated articles
Author Response
In this review, the authors described the role of autophagy in cancer as well as the potential agents that exert anticancer activity by regulation of autophagy. This was accompanied by summary of autophagy’s functions in cancer cells and explanation of potential anticancer drugs that modulate autophagy for therapeutic targeting in cancer. There are minor corrections and suggestions to improve the quality of the manuscript, and these are listed as in the following:
Authors should improve the formatting of the manuscript
Thank you for the comments. We solved the formatting issue
2. Authors should revise the references that use the more recently updated articles
we revised the references eliminating the old ones and the redudancy.